# Evaluation of Cua Lo Estuary's Morpho-Dynamic Evolution and Its Impact on Port Planning

Dinh Nhat Quang [1], Nguyen Quang Duc Anh [2], Ho Sy Tam [1,*], Nguyen Xuan Tinh [3], Hitoshi Tanaka [4] and Nguyen Trung Viet [1]

1    Faculty of Civil Engineering, Thuyloi University, 175 Tay Son, Dong Da, Hanoi 10000, Vietnam
2    Institute of Civil Engineering, Thuyloi University, 175 Tay Son, Dong Da, Hanoi 10000, Vietnam
3    Water Resources Management Division, CTI Engineering International Co., Ltd., 2-25-14 Kameido, Tokyo 136-0071, Japan
4    Institute of Liberal Arts and Sciences, Tohoku University, Sendai 980-8576, Japan
*    Correspondence: tamhs.cttl@tlu.edu.vn

**Abstract:** Coastal erosion and accretion along the Quang Nam coast in Vietnam have been increasing in recent years, causing negative impacts on the inhabitants and local ecology. The Cua Lo estuary in Nui Thanh district has a complex hydrodynamic regime owing to its connection with two estuaries and three different tributaries. Therefore, a detailed study of the mechanisms and processes of these phenomena is crucial to understand the potential impact of a proposed 50,000-ton cargo port. In this study, the Delft3D model is employed to evaluate the morpho-dynamic changes in the area of Cua Lo under monsoon wave climate, storm, and flood conditions both before and after port and navigation channel construction. Results indicate that in the absence of the port, tidal currents and waves during monsoon storms cause significant erosion on the south bank and accretion on the north bank. Furthermore, the GenCade model is utilized to predict the future shoreline changes after the construction of two jetties. The model reveals that after 50 years of operation, the shoreline modifications will extend 449 m towards the sea, in comparison to natural conditions. However, the design of the northern jetty will ensure safe and proper operation without impacting the navigation channel. This study offers valuable insights into the morphological changes in the Cua Lo area and their potential implications, which can aid in the development of sustainable coastal management strategies for the region.

**Keywords:** Cua Lo estuary; Delft3D; GenCade; morpho-dynamic evolution; Quang Nam province; sediment transport

## 1. Introduction

Coastal area is favorable for economic development with opportunities for transportation, industry, fishing and aquaculture, recreation, tourism, etc. It brings great benefits and revenue both in terms of economy and politics for countries [1,2]. However, coastal areas also face challenges such as vulnerability to natural disasters like hurricanes and flooding, which can damage infrastructure and disrupt economic activities [3]. Additionally, the phenomenon of coastal erosion and accretion recently has resulted in adverse effects such as environmental degradation, impaired flood drainage capabilities, risks to life and property, loss of natural habitats and tourism revenue, and negative impacts on the local community [4].

Quang Nam province, located in the central part of Vietnam, boasts a long coastline that features a rich ecological environment and beautiful creatures [5,6] (Figure 1). The long coastline has a positive influence on the development of transportation, economy, and tourism, which allows local people to make a good living and contributes to the socio-economic development of Quang Nam province [7,8]. The Cua Lo area, located in Nui Thanh district, offers excellent development potential thanks to its unique geographical

location near major economic regions, as well as Chu Lai airport and Ky Ha and Tam Hiep ports. Moreover, An Hoa lagoon, located deep within Cua Lo and Ky Ha estuaries, is an ideal location for a fishing port and storm shelter for fishing boats operating in the in storm-prone region. The An Hoa fishing port in Tam Giang commune is a grade-II fishing port with an annual seafood output of 16,000 tons.

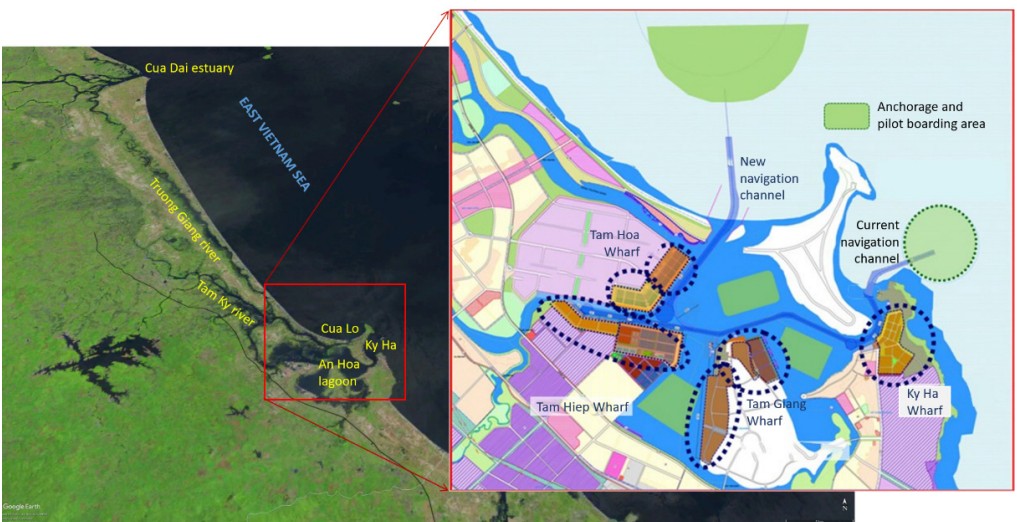

**Figure 1.** Location of the study area and Master plan for Chu Lai port.

Quang Nam province has put into operation Chu Lai Port as a key logistics hub for the central and Central Highlands region (e.g., connect with industrial parks and economic zones in Central Vietnam and Central Highlands, South Laos, North Cambodia) and an international shipping hub (i.e., a gateway to the East sea for Northeast Asia). Currently, Chu Lai port is capable of receiving 50,000-ton ships with its modern large warehouse system that is meeting European standards convenient for storing, loading, and unloading cargo (Figure 1). Presently, there are two navigation routes to the port, i.e., through Cua Lo and Ky Ha estuaries, while the route through Ky Ha only has capacity to receive 20,000-ton cargo ships. Therefore, there is the requirement for upgrading the maritime route that is able to receive 50,000-ton cargo ships. Taking into account the harmony with nature and avoidance of the shallow rocky reef, a new navigation channel through Cua Lo had been proposed with the length of 6 km and the width of 140 m (Figure 1). Two jetties had been designed to protect the channel from large waves and to prevent sediment deposition within the channel, thus guaranteeing the safety of the cargo ships. Two design alternatives for the jetties, namely OPT1 and OPT2, are considered (see Figure 2), in which the southern jetty in OPT1 alternative does not block the flow from Truong Giang and Tam Ky rivers to the sea through Cua Lo inlet. In contrast, the southern jetty in OTP2 alternative totally blocks the flow to Cua Lo inlet.

Quang et al. [7] investigated the long-term spatio-temporal shoreline evolution along the Quang Nam coast using satellite images over 30 years (1990–2019) by DSAS software. The authors stated that the average coastline recession/accretion rates for the Quang Nam coast obtained by two parameter functions in DSAS, i.e., End Point Rate (EPR) and Linear Regression Rate (LRR), are −1.7/0.77 m/year and −1.86/0.83 m/year, respectively. With both sides of Cua Lo estuary being sandy shores, the influence of the river network combined with the seasonally varying wave and flow regimes has created a highly dynamic morphological system. More precisely, the sand spit situated on the left bank of Cua Lo inlet had extended approximately 1.7 km to the right over period 1973-2018 (Figure 3). Since the southern coast of Cua Lo is shielded by Ban Than rocky spit and coral reefs—which act as the no-transport boundary, the sediment transported along the coast from Cua Dai to Cua Lo accumulates continuously in the northern coast of Cua Lo and results in the elongation of the sand spit. Consequently, the flow from Truong Giang and Tam Ky river systems in

combination with the ebb tide cause strong erosion on the right shore of Cua Lo. These phenomena not only affect the navigation of cargo ships and fishing boats but also reduce flood drainage capacity from Truong Giang and Tam Ky rivers [9,10]. Thus, this area is considered the "hot-spot" of accelerated erosion issues, having detrimental influences on the sustainable management and development of the region.

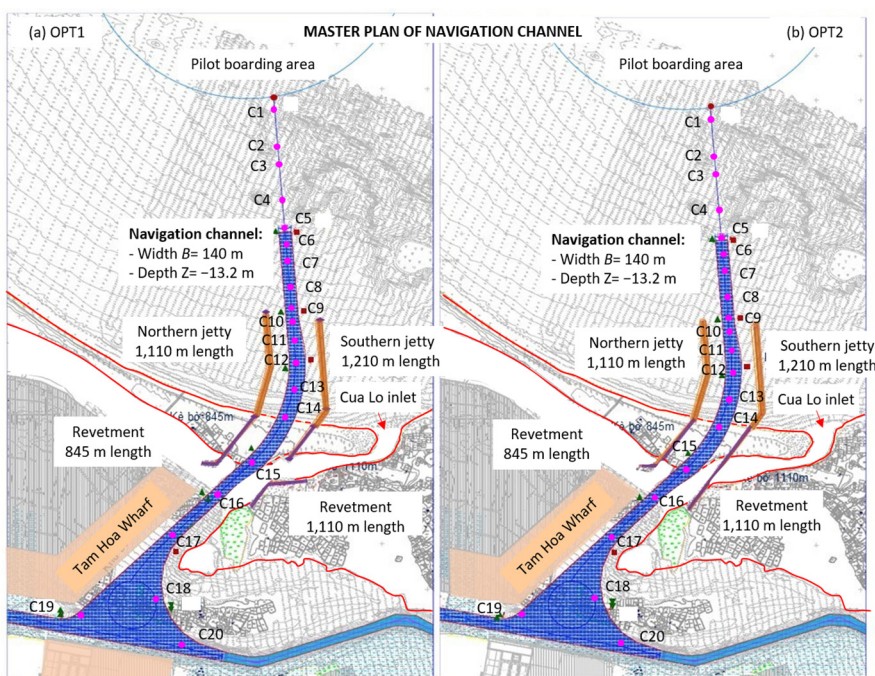

**Figure 2.** Master plan of navigation channel with two design alternatives of jetties (**a**) OPT1 Plan, (**b**) OPT2 Plan.

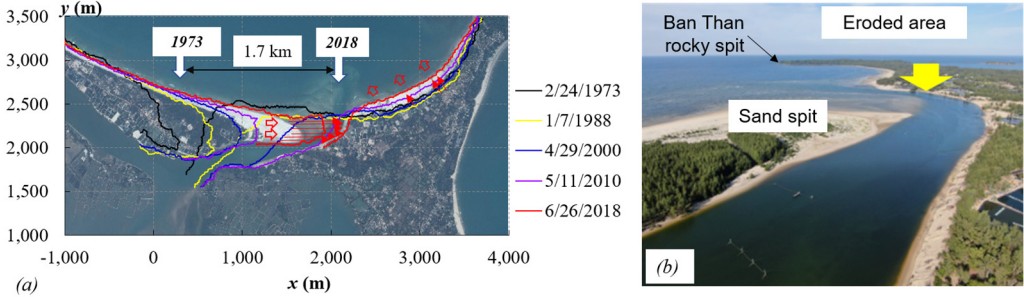

**Figure 3.** (**a**) Cua Lo sand spit elongation over period 1973–2018, (**b**) location of the elongated sand spit and eroded area.

This study employs the Delft3D model to evaluate the hydrodynamic and morphological changes in the area of Cua Lo under the impact of the monsoon wave climate, typical storm, and river flood conditions, both in the current situation and after the construction of navigation channel. Additionally, the GenCade model is adopted to investigate the impact of jetty construction on shoreline change along the sandy coast of Quang Nam after several years of operation. The adoption of these models aims at gaining a deeper understanding of the mechanisms and processes that drive coastal erosion and accretion in this region and investigating the potential impact of a proposed 50,000-ton cargo port. Key evaluation criteria include flood discharge capacity, wave reduction ability, and amount of deposited sedimentation within the navigation channel.

## 2. Materials and Methods

### 2.1. Materials

For numerical modeling purposes, the following data are attempted to be collected.

### 2.1.1. Satellite Images

In this research, Landsat 5 and Landsat 8 images are used for the development, calibration, and validation of the models. More precisely, only the satellite images in March with the cloud presence less than 10% are selected to avoid negative natural phenomena, i.e., storms, typical in the rainy season when the wave energy is stronger and causes the threat of coastal erosion temporarily (Table 1).

**Table 1.** Selected satellite images.

| Nº | Date | Satellite | Path/Row |
| --- | --- | --- | --- |
| 1 | 10 March 2005 | Landsat 5 | 124/049 |
| 2 | 24 March 2010 | Landsat 5 | 124/049 |
| 3 | 6 March 2015 | Landsat 8 | 124/049 |

### 2.1.2. Topography and Bathymetry Data

Topography map with the scale of 1/10,000 for the study area is collected from Department of Survey, Mapping and Geographic Information Vietnam–Ministry of Natural Resources and Environment. Bathymetry data of Cua Lo estuary and its adjacent area with the scale of 1/5000 and 1/2000 surveyed in 2019 and 2020 are collected from the bilateral project between Vietnam and Japan [10] (Figure 4). In addition, the bathymetry data for offshore area is obtained from GEBCO's gridded bathymetric data sets [9,11].

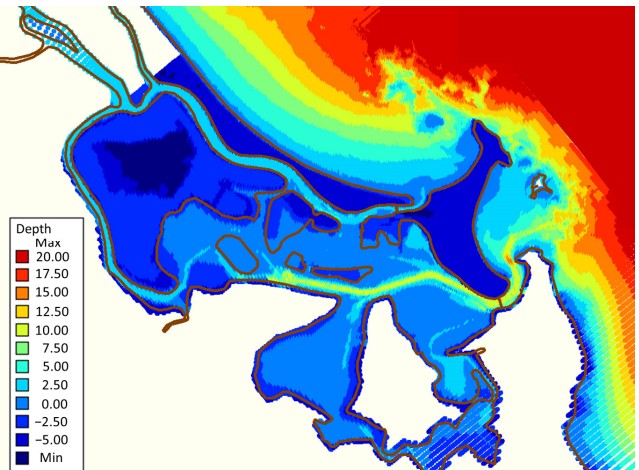

**Figure 4.** Bathymetry of Cua Lo estuary and its adjacent area in 2019.

### 2.1.3. Wave and Water Level Data

The wave data used in this research are extracted from the wave model for the South China Sea, which was developed by Dien et al. [12] and covers the period from 2000 to 2020. The wave model was developed by using SWAN model and well-calibrated and validated using measured tidal levels at gauging stations within the study area. The wave data are extracted at Point P (108.75° longitude; 15.75° latitude), as shown in Figure 5, revealing that the wave regime in Quang Nam can be classified into two distinct monsoon seasons, i.e., winter and summer monsoons. During the winter monsoon, the dominant waves are in the Northeast, East-Northeast, and East directions, with corresponding frequencies of 14.35%, 33.42%, and 11.78%, respectively. In contrast, during the summer monsoon, the dominant waves are in the Southeast and South-Southeast directions, with frequencies of 22.14% and 11.86%, respectively.

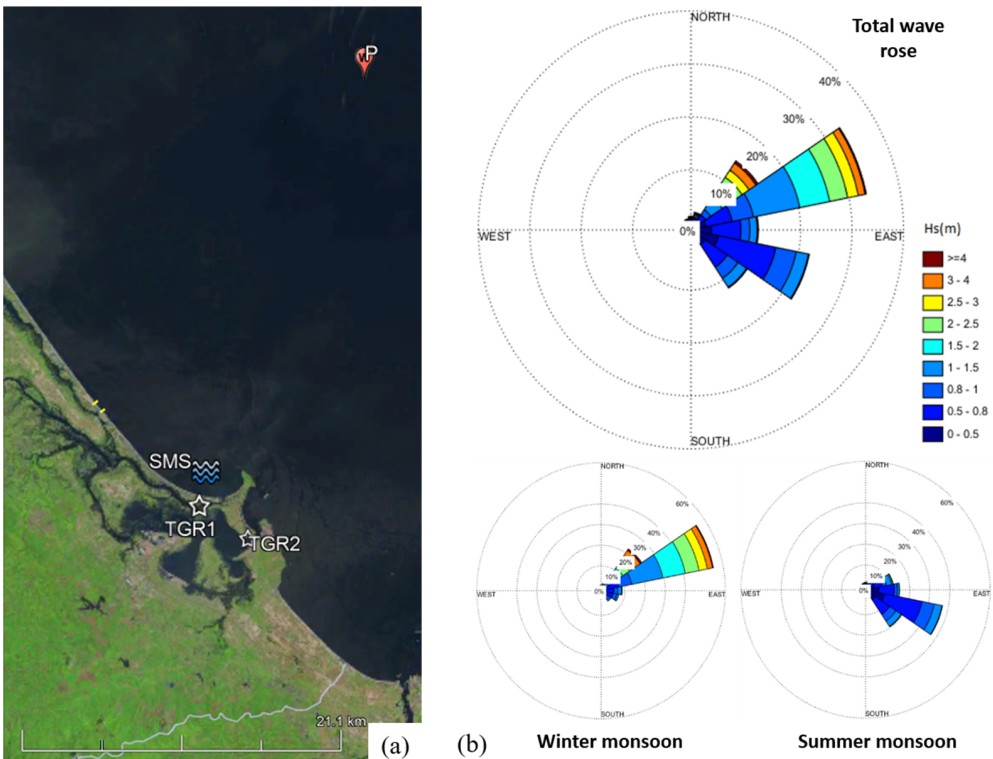

**Figure 5.** (**a**) Locations of water level and wave stations, (**b**) wave roses extracted at Point P.

For the purpose of calibrating and validating of the hydrodynamic model developed for the Quang Nam coastal area, the water levels and offshore wave data are collected at stations TGR1 (108.647°; 15.494°), TGR2 (108.677°; 15.474°), and SMS (108.651°; 15.505°) from two survey campaigns in 11–18 August and 12–21 November 2019. At the SMS station, wave characteristics such as wave height, wave period, and wave direction were monitored using an Acoustic Wave and Current Profiler system. These data are crucial for accurately calibrating and validating the hydrodynamic model for the study area.

### 2.1.4. Sediment Data and Beach Characteristics

The significant long-shore sediment transport from Cua Dai estuary to Cua Lo estuary serves as the primary source of sand for the elongation of Cua Lo sand spit. Utilizing remote sensing analysis and mathematical modeling, Anh [13] estimated the annual sediment transport rate along the coast of Cua Lo area to be approximately $0.8–1.6 \times 10^5$ m$^3$/year.

Data on grain size and physical properties of the sediment within the study area are collected from the project of Tam et al. [10]. More specifically, more than 100 bottom sediment samples had been collected in the area of Cua Lo estuary during the survey campaign in August 2019, with the median particle diameter (D50) of 0.2 mm. Duy et al. [14] estimated the seaward limit of significant sediment transport called the depth-of-closure $D_c$—a vitally important parameter in the shoreline change numerical model is 6 m. Mau et al. [15] estimated the berm height $D_b$ within the study area is 2 m.

### 2.2. Methodologies

The process-based numerical model suite Delft3D was developed by Delft Hydraulics Research Institute and widely applied to assess the coastal morphological changes [16]. This model allows simulating wave motion, flow, and sediment transport and links these components in the morphological simulation problems by several types of modules in its system [17,18]. In particular, the variation of wind-generated waves is modeled by Delft3D-WAVE module, which is built based on standard wave models, like HISWA (Hindcast Shallow Water Waves) and SWAN (Simulation Waves Nearshore) [19,20]. The calculated

wave parameters in Delft3D-WAVE model are often used as input for Delft3D-FLOW module, during wave propagation simulation [21]. When the wave breaking limit is determined, several typical parameters are calculated, i.e., wave-driven currents, enhanced turbulence, bed-shear stress, and sediment stirring. The model suite has been widely used to simulate hydrodynamics and morpho-dynamics in estuary areas. For example, Delft3D was used to simulate the hydrodynamic and sediment transport at the mouth of the Columbia River [22,23]. Boudet [24] investigated sediment transport patterns in the mouth of Rhone Delta under different wave and flow conditions during flood and storm events, which are the major morphological drivers of this region. Fernández et al. [25] developed Delft3D model for the Mondego River estuary system, a jettied mouth wave-dominated ebb-tidal inlet at Figueira da Foz, to simulate the morphological evolution of five dredging scenarios under three wave climate scenarios.

Mathematical modeling is the most useful method for the prediction of coastal change at all time and space scales [26]. Among the shoreline change models, the one-line model is commonly used to evaluate and predict the evolution of the long-term shoreline change, including the effects of constructions and other measures on the coastline [27]. Anastasiou and Sylaios [28] predicted the shoreline change in the Nestos River Delta, Greece, using LITPACK model and stated that the model showed fair agreement to remote sensing analysis over 31 years. Yan Ding et al. [29] used a newly developed one-line model GenCade to simulate shoreline changes along the Delaware Coast and showed a good agreement between simulated and observed shorelines. Sung-Chan Kim et al. [30] adopted and compared two wide-used one-line models, i.e., LITPACK and GenCade, through a series of simplified test cases to the analytical solution. The results reveal that both models give good agreement with the analytical solutions. However, the GenCade model allows for user-specified, time-series beach placements starting after the simulation start time and assuming wave diffraction for groins, thereby helping the results from GenCade give more accurate results in predicting shoreline change. Townsend et al. [31] used three one-line models, i.e., LITPACK, UNIBEST, and GenCade, to evaluate the evolution of shoreline position. The authors stated that the GenCade model has internal models of wave diffraction, and an inlet bypassing capability and an inlet reservoir model for the calculation of shoal and inlet feature sediment balance.

In this research, the Delft3D model is adopted to evaluate the hydrodynamic and morphological changes in the area of Cua Lo under the influence of the monsoon wave climate, storm, and flood conditions, while GenCade model is used to assess the shoreline evolutionary trend along the Quang Nam coast after jetty construction.

2.2.1. Development of Hydrodynamic and Sediment Transport Model for the Area of Quang Nam Coast and Cua Lo Estuary

- Delft3D model setup

The study area is a coastal estuary with highly intricate topography and wave characteristics that are main driving forces of the sediment mobilization in the nearshore zone. More precisely, the northern coast is protected by Cu Lao Cham Island and affected by Son Tra Peninsula, whereas along the southern coast, the wave regime is strongly influenced by Dung Quat Bay (Figure 6). Moreover, Cua Lo area is shielded by Ban Than cape and coral reefs that act as a natural breakwater (see Figures 1 and 3). Cua Lo estuary is also connected to Ky Ha estuary through An Hoa lagoon and Truong Giang, Tam Ky river systems. Therefore, the computational domain has been meticulously established.

A coupled model has been developed for the study area, incorporating both the wave and flow models. The computational domain of the coupled model consists of two nested numerical grids generated by Delft3D-RGFGRID module (Figure 6). The larger domain is established for the wave model (Delft3D-WAVE module), covering nearly 80 km from the north shore of Cua Dai estuary to the end of Dung Quat Bay with the area of 3000 km$^2$. This domain consists of $182 \times 59$ grid cells with the sizes varying from 300 m to 600 m. As the wave enters shallower regions, it is profoundly influenced by local bathymetry and

geometry. Therefore, the finer model domain is established for the flow model (Delft3D-FLOW module) surrounding the area of Cua Lo, including the Truong Giang and Tam Ky river systems, in order to better determine the actual local wave characteristics as well as fully consider the interaction between the river and the sea. The finer domain encompasses an area of approximately 500 km², comprising 287 × 323 grid cells, with the smallest size of 10 m (e.g., within the estuary area, along the coast and in An Hoa lagoon), and their sizes gradually increase toward the sea.

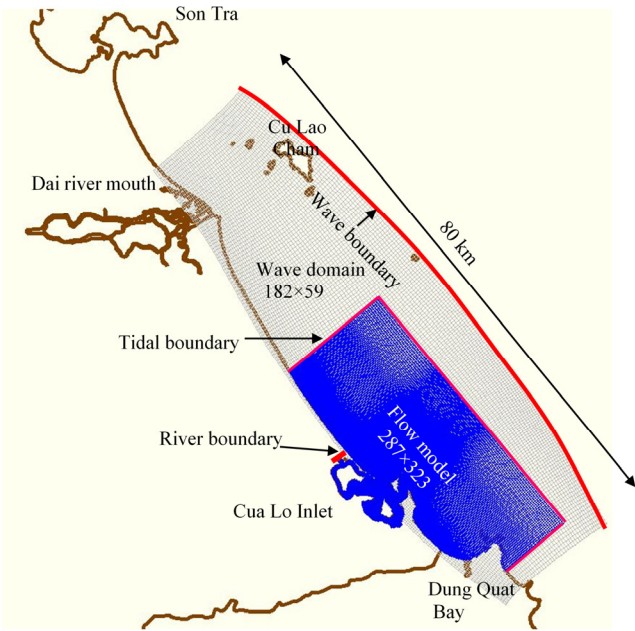

**Figure 6.** Wave and flow model domains and grid resolutions.

Boundary conditions of the coupled model include wave boundary, tide boundary, and river boundary. The wave parameters, i.e., wave height $H_s$, wave direction $\theta$, and wave period $T_p$ are extracted at Point P (see Section 2.1.3). The tidal boundary is established through the harmonic constants, i.e., 8 main components $M_2$, $S_2$, $N_2$, $K_2$, $K_1$, $O_1$, $P_1$, and $Q_1$ with the spatial resolution of 0.5°, obtained from the global tidal database TPXO [32], by using Delft3D Dashboard. The river boundary, i.e., river hydrograph, is obtained from the hydraulic model for the entire Truong Giang and Tam Ky river systems developed by Tam et al. [10] and Quang et al. [33]. The initial numerical bathymetry is built with Delft3D-QUICKIN using bathymetry data described in Section 2.1.2 (Figure 7).

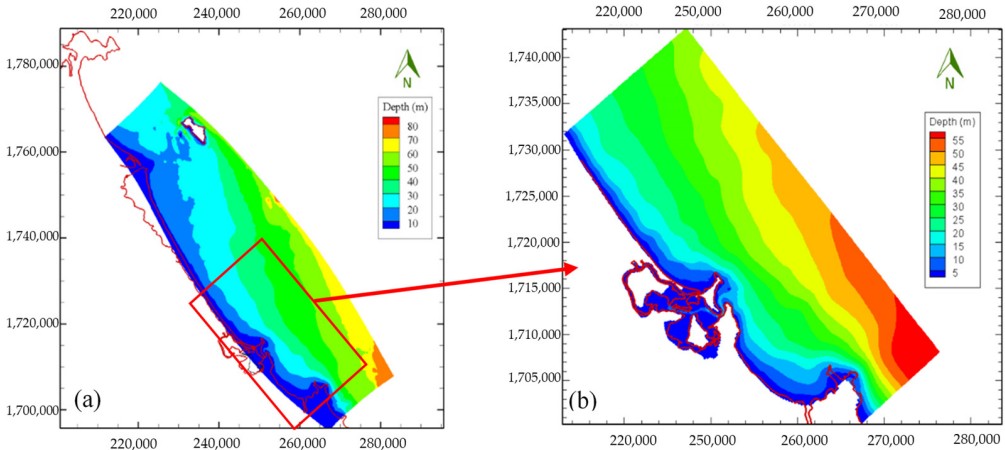

**Figure 7.** Initial numerical bathymetry for (**a**) large domain, (**b**) finer domain.

- Delft3D calibration and validation

The calibration and validation processes of the coupled model are performed by using the surveyed wave and water level data during the summer (in August 2019) and winter (in November 2019) periods (see Section 2.1.3). Two statistical parameters, i.e., Root Mean Square Error (RMSE) and Mean Error (ME), are adopted to evaluate the performance and reliability of the model. Figure 8 shows that the simulated water level and wave height fit quite well with the observed data in terms of both magnitude and phase. The wave model satisfactorily captures the time variation of wave height during the Northeast monsoon in November 2019. Even though the time variations of wave height in August 2019 survey are not reproduced well, the model results are still satisfactory since the obtained RMSE and ME values are sufficiently small. The RSME values for the water level at TGR1, TGR2 stations and wave heights at SMS station are 0.04 m, 0.05 m, 0.06 m, and 0.11 m, respectively, while their ME values are 0.03 m, 0.04 m, 0.045 m, and 0.05 m, respectively. Thus, this well-calibrated and validated coupled model is reliable to be adopted for next simulation steps.

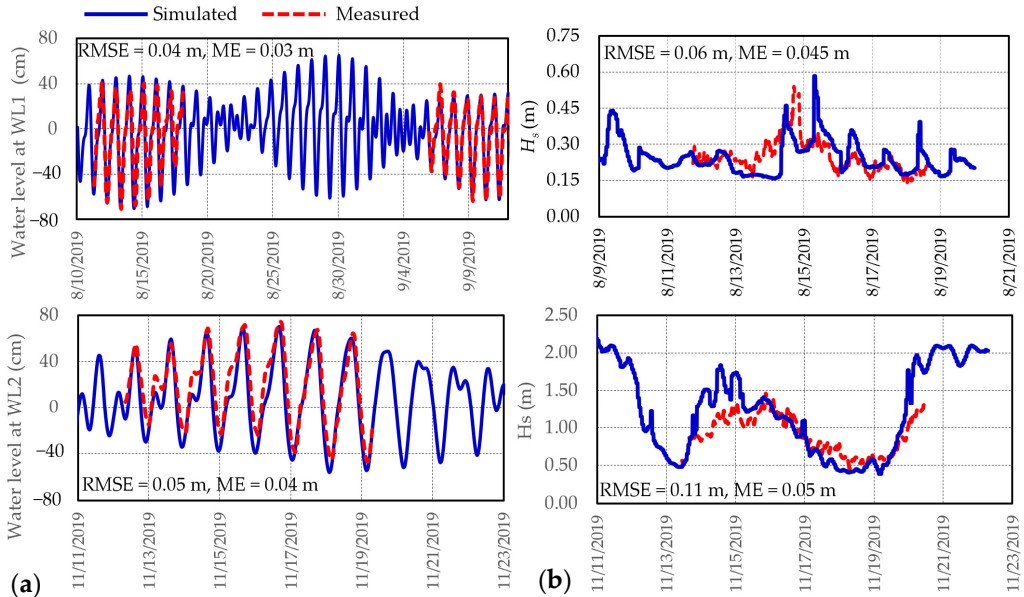

**Figure 8.** Comparison between (**a**) simulated and observed water levels at TGR1 and TGR2 stations, (**b**) wave heights at SMS station.

- Model scenarios

In this study, the hydrodynamic regime and sediment transport under different scenarios based on the influence of monsoon wave climate, storm, and river flood conditions are investigated in order to clarify and compare the hydrodynamic and morphological changes under two conditions of hard structure, i.e., current situation and after jetties construction (see Table 2).

**Table 2.** Simulation scenarios.

| Scenarios | Current Situation (*BL*) | Jetties Construction (OPT1, OPT2) |
|---|---|---|
| Northeast monsoon period | x | x |
| Southeast monsoon period | x | |
| Typical storm | x | x |
| Typical flood | x | x |

### 2.2.2. Development of One-Line Shoreline Model for Quang Nam Coast

GenCade is a newly developed model for calculating sediment transport and morphology change along coastal regions [34]. This is synthesized and combined with the GENESIS model and the CASCADE model [35,36]. GenCade is developed and maintained by the U.S. Army Engineer Research and Development Center, Coastal and Hydraulics Laboratory with support from the Coastal Inlets Research Program and the Regional Sediment Management program.

- GenCade model setup

Figure 9 illustrates the main steps to set up and run the GenCade model. Firstly, the input conditions and parameters need to be defined and assigned (Figure 10). The initial coastline from Cua Dai estuary to Cua Lo estuary (green line) is extracted from Landsat 5 image on 10 March 2005 by applying the method proposed by Quang et al. [6,7]. The coastline from Cua Dai estuary to Cua Lo estuary is divided into 3124 cells (black line) with the size of 15 m. The offshore wave boundary conditions are introduced at four points, i.e., $P_1$ (108.485°; 15.903°), $P_2$ (108.564°; 15.777°), $P_3$ (108.635°; 15.675°), and $P_4$ (108.709°; 15.581°), at the water depth of 30 m; those are extracted from the wave model for the area of Quang Nam coast for period 2000–2020. Sand and beach data are assigned with the values of 0.2 mm, 2 m, and 6 m for the effective grain size, berm height $D_b$, and closure depth $D_c$, respectively (see Section 2.1.4). The choice of longshore sand transport calibration coefficients $K_1$ and $K_2$ is described hereafter.

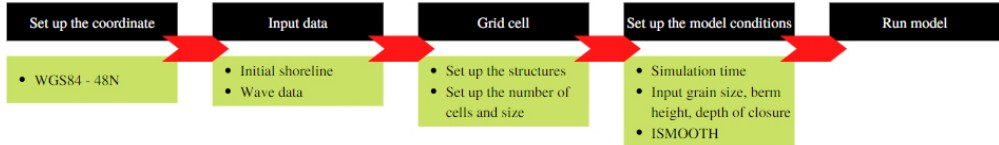

**Figure 9.** Main steps for running the GenCade model.

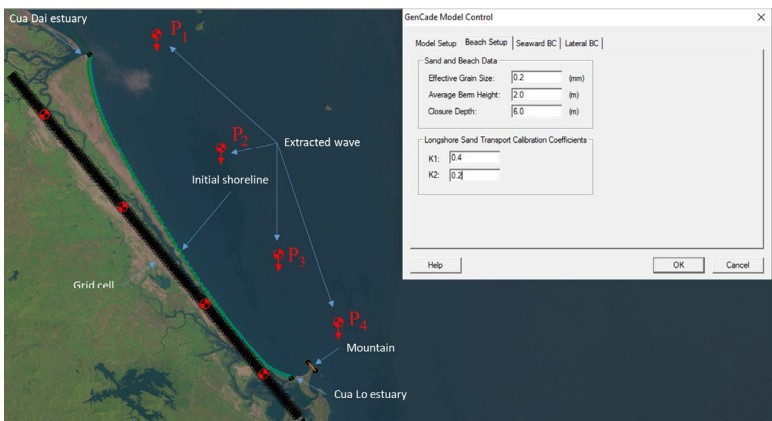

**Figure 10.** Establishment of the GenCade model from Cua Dai estuary to Cua Lo estuary.

- Selection of the most suitable values of $K_1$ and $K_2$

The empirical coefficients $K_1$ and $K_2$ in GenCade model play a crucial role in evaluating and predicting the shoreline change in the long-term period [34]. Based on modeling experience, Hanson and Kraus [35] recommended $0 < K_1 < 1$ and $0.5 K_1 < K_2 < 1.5 K_1$. In this study, 30 pairs of empirical coefficients $K_1$ and $K_2$ (red circles) have been tested to select the most suitable values of $K_1$ and $K_2$ for the study area by comparing the simulated shorelines with the actual shoreline. More precisely, GenCade model simulates the shoreline change from 2005 to 2010 using the "*Initial shoreline*" in 2005 (green line in Figure 10); then, its simulated shoreline is compared with the actual shoreline extracted from the Landsat 5 image on 24 March 2010. The results reveal that the most suitable coefficients $K_1$ and $K_2$ for

the Quang Nam coast are 0.4 and 0.2, with the values of MAE *(Mean Absolute Error)* and RMSE being 2.39 m and 3.22 m, respectively (Figure 11).

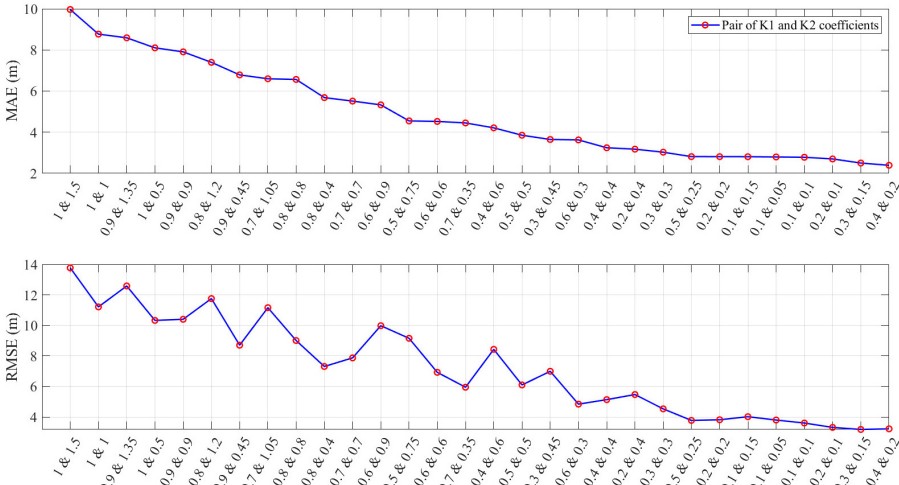

**Figure 11.** Performance of the one-line model based on two indices: MAE (**upper** panel), RMSE (**lower** panel).

The one-line model is validated by simulating the shoreline change for the period 2010–2015 and compared to the actual shoreline extracted from Landsat 8 image on March 2015. The model still performs well in predicting the accretion and erosion trends of the shoreline with the values of MAE and RMSE being 2.40 m and 3.41 m, respectively. Therefore, this well-calibrated and validated model with optimal couple of $K_1$ and $K_2$ will be adopted to evaluate the shoreline change along Quang Nam coast under the construction of two jetties.

## 3. Hydrodynamic and Morphological Changes in the Area of Cua Lo Estuary under the Natural Condition

### 3.1. Hydrodynamic and Morphological Changes under the Impacts of Monsoons

The wave regime within the study area is mainly influenced by the propagation of deep-water waves into the coastal area and can be classified into two distinct monsoon seasons, i.e., winter monsoon (from October to March, the dominant wave direction is in the Northeast) and summer monsoon (from April to September, the dominant wave direction is in the Southeast). Figure 5 shows that the Northeast monsoon dominates the Southeast monsoon both in term of frequency and energy. Based on the statistical analysis of the wave series at Point P, the values of significant wave height, wave period, and wave direction are 2.96 m, 9.56 s, and 50° for the Northeast monsoon and 1.07 m, 6.32 s, and 112° for the Southeast monsoon, respectively.

Figure 12 shows the results of wave height during the Northeast and Southeast monsoons under current condition (*BL—Baseline*). In the Northeast monsoon, the wave distribution in the inlet area and its Northern and Southern sides are markedly different. More specifically, the wave height in the Northern part of Cua Lo area can reach up to 2 m before the wave breaks, while at the inlet, the wave height ranges from 1.0 m to 1.5 m. In the Southern part of Cua Lo estuary, Ban Than cape and coral reefs have a great influence on wave propagation where the wave height is under 0.5 m, and the waves break quite far from the shore. During the Southeast monsoon, the wave height is relatively small compared to the Northeast monsoon period, e.g., around 0.2–0.4 m in the area of Cua Lo. The study area is shielded by Ban Than cape and coral reefs, so it is quite calm during the Southeast monsoon period for all incoming wave scenarios.

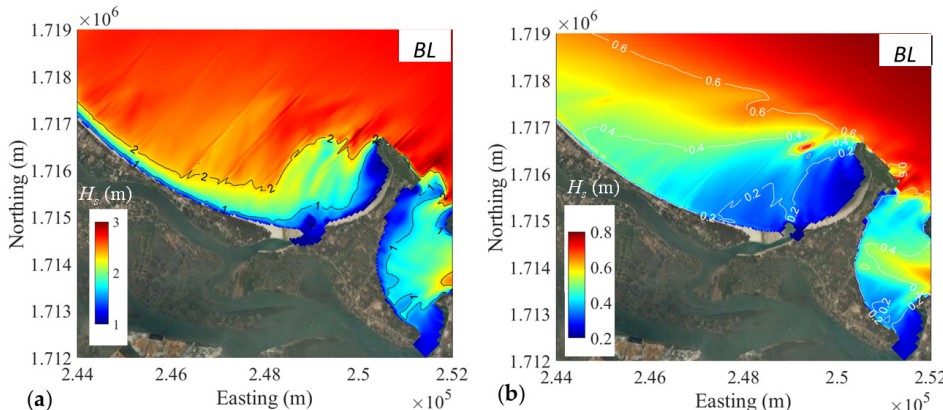

**Figure 12.** Wave height during (**a**) Northeast monsoon and (**b**) Southeast monsoon under current conditions.

Figure 13 shows the simulated current field in the flood tide (tide rises) and ebb tide (tide recedes) phases during the Northeast monsoon. In both tidal phases, the area with high velocity (red color) appears on the right side of the estuary. This is the cause of serious erosion in Tan Lap and Thuan An villages and Tam Hai commune as shown in Figure 3. Moreover, the combined currents under the action of waves during the Northeast monsoon create residual currents parallel to the shoreline and oriented from South to North. Consequently, a counter-clockwise circulation is formed, taking the sand from the outer sand bar to accrete to the tip of the sand spit in the North.

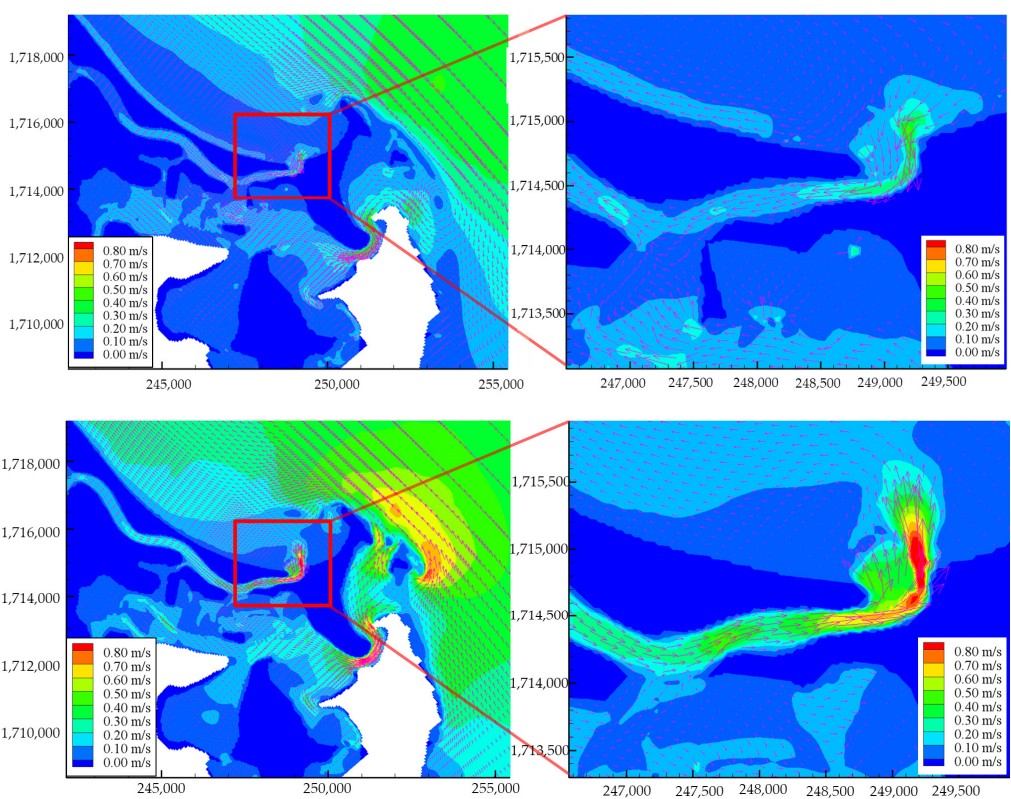

**Figure 13.** Current field in the flood tide (**upper** panel) and ebb tide (**lower** panel) during the Northeast monsoon under current condition.

In order to evaluate the characteristics of bed level changes, the simulation for a typical year under the Baseline scenario with the initial bed level in 2019 is conducted. Figure 14 shows the bed level changes in the area of Cua Lo estuary under Baseline scenario, where

the warm color (red) represents the accretion areas, and the cold color (blue) represents the erosion area. Offshore red alluvial areas can be seen along the north shore of Cua Lo inlet. This is due to the predominant cross-shore sediment transport under the Northeast waves with high energy level. Under current situation, the sediment is transported from the river under the impact of the inlet current, combined with the sediment flow transportation by wave circulation accreted at the berm area. The simulation results also reveal that the Northern sand-spit is formed and developed towards the South, while the erosion area is formed in the southern area of Cua Lo. The area behind Ban Than cape and coral reefs is stable due to significant dispersion of wave energy. The results of bed level changes are consistent with the hydrodynamic mechanism presented in Figures 12 and 13.

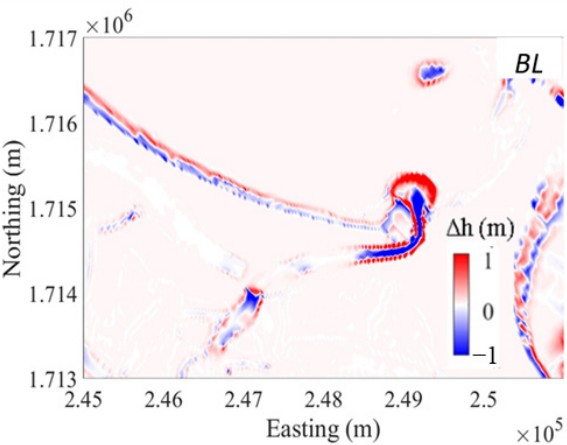

**Figure 14.** Bed level changes in the area of Cua Lo estuary under current condition.

### 3.2. Hydrodynamic Changes under the Impact of Storm

According to storm data from Japanese Meteorological Agency, the coastal area of Quang Nam province has been affected by 79 storms between 1951 and 2019. The majority of strong storm events occurred in September (30.4%), followed by October (27.8%), and November (13.9%). In this study, typhoon Molave, which hit the Central Coast of Vietnam in late October 2020 and was the strongest typhoon to strike this area since typhoon Damrey in 2017, has been chosen to assess the impact of storm on the hydrodynamic regime and sediment transport.

Figure 15a shows the wave field of the study area during typhoon Molave under the current situation (BL). The typhoon arrived at the low tide phase with tidal level amplitude ranging from −0.3 to 0.3 m. The effects of typhoon on water level fluctuations are clearly seen; in particular, the water level rose over 0.2 m (i.e., the water level rises due to the impact of the wind within the study area, excluding other storm surge components). The wave height reaches 5 m at Ban Than cape and decreases rapidly due to the topographical characteristics. In the area of Cua Lo, the transmitted waves are significantly reduced due to the shielding of the coral reef and Ban Than cape. Figure 15b shows the maximum flow velocity during typhoon Molave and reveals that the flows in the sea and near the shore tend to be larger due to the impact of large waves and winds during the typhoon.

### 3.3. Hydrodynamic and Mophorlogical Changes under the Impact of River Flood

Figure 16a shows the maximum current field during the flood event in November 2017, which has the frequency of 5% [33]. The flood flow from Truong Giang and Tam Ky river systems pours into An Hoa lagoon and enters the sea through Cua Lo and Ky Ha estuaries (Figure 1). According to Tam et al. [10], the maximum discharge through Cua Lo estuary during the peak flood period in November 2017 is about 550 $m^3/s$; meanwhile, its average discharge under normal conditions is about 450 $m^3/s$. Under the flood condition, the maximum velocity at Cua Lo reaches up to 1.5 m/s on the right side of Tam Hai commune. Due to this high velocity, serious erosion on the right bank of Cua Lo occurs.

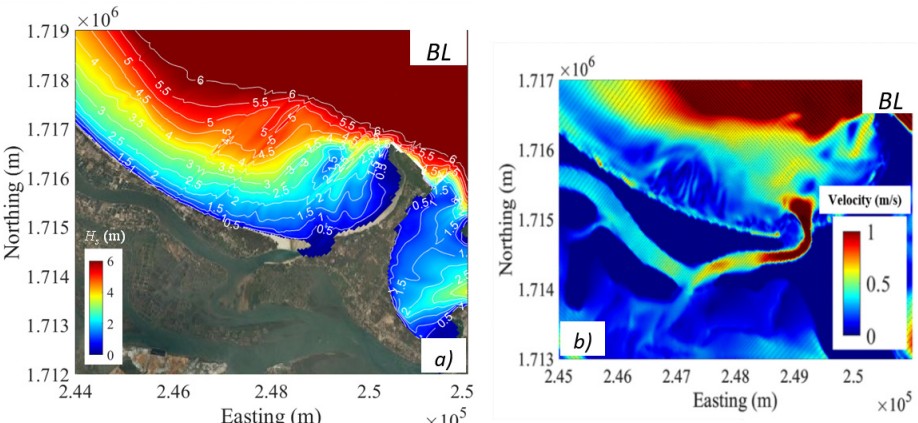

**Figure 15.** (**a**) Wave field, (**b**) current field under the impact of typhoon Molave—BL scenario.

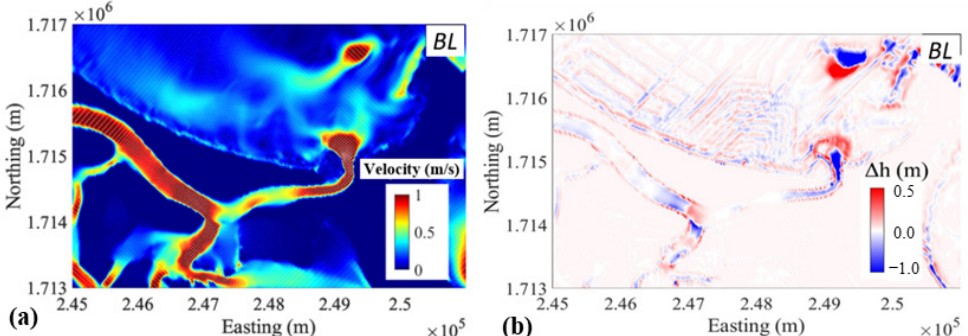

**Figure 16.** (**a**) Current field, (**b**) bed level changes in the area of Cua Lo after the flood event—BL scenario.

Figure 16b shows the bed level changes in the area of Cua Lo after the flood event in November 2017. There are alternating accretion and erosion areas along the Truong Giang river to Cua Lo inlet. The flood flow causes serious erosion in the area on the right bank of Cua Lo with the maximum depth of 0.5 m. This large amount of eroded sediment is the source for forming the sand spit that elongates toward the estuary.

## 4. Impact Assessment of the Port Planning and Regulation Work Construction

A comparative study has been conducted to evaluate the effectiveness of two construction alternatives OPT1 and OPT2 based on three key evaluation criteria, i.e., flood discharge capacity, wave reduction ability, and amount of deposited sedimentation within the navigation channel during port operation.

### 4.1. Assessment the Impact of Jetty Construction on Hydrodynamic Regime and Sediment Transport

4.1.1. Impact Assessment of Jetty Construction during the Monsoon Season

Figure 17 shows the largest wave field during the Northeast monsoon in the area of Cua Lo after the construction of the navigation channel according to OPT1 and OPT2 alternatives. The construction has changed the wave distribution compared to natural conditions, where the wave height between two jetties decreases from 1–2 m to only 0.25–1 m. However, the wave height behind the crest of the Southern jetty is still relatively high, i.e., of about 2 m.

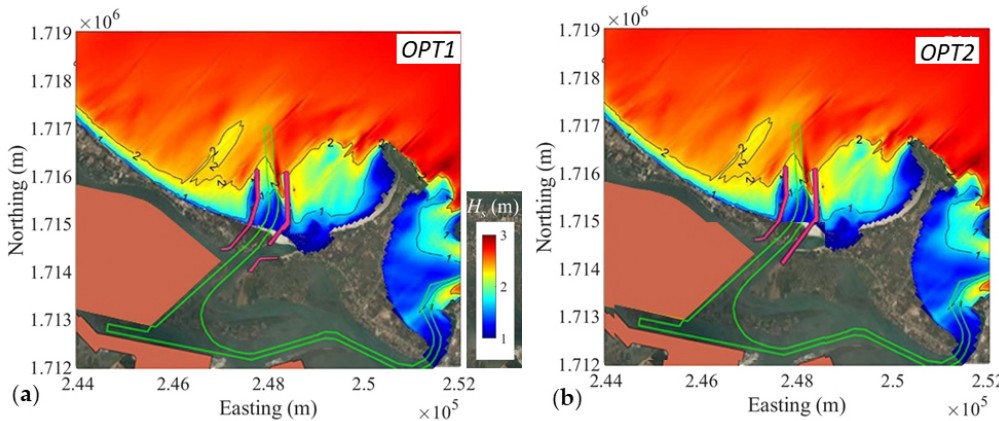

**Figure 17.** Wave field during the Northeast monsoon after jetty construction.

Figure 18 illustrates the current field during the Northeast monsoon after jetty construction. Under the OPT1 alternative, the flow mainly concentrates through the new navigation channel with the maximum velocity of being 0.5–0.7 m/s. Although the current Cua Lo inlet is closed under the OPT2 alternative, the flow velocity within the channel is not significantly different from the one under OPT1 alternative, of around 0.5–0.8 m/s. Figure 19 shows the bed level changes in one year after jetty construction. Due to the flow field contribution, the bed level change is mainly concentrated in the channel area.

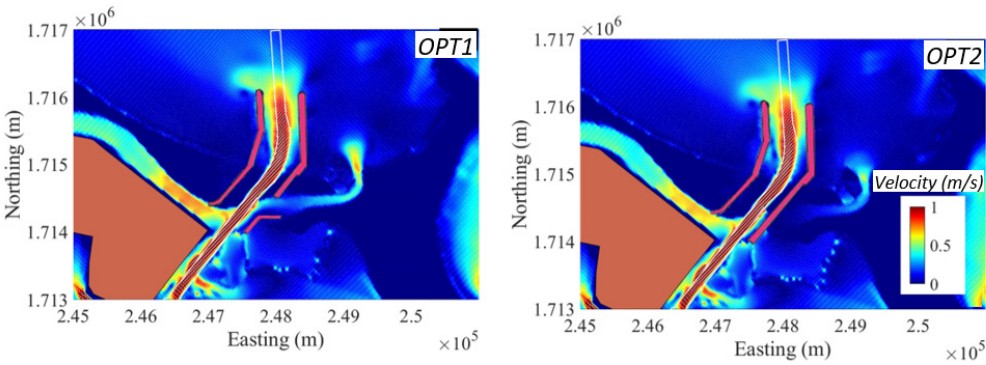

**Figure 18.** Current field during the Northeast monsoon after jetty construction.

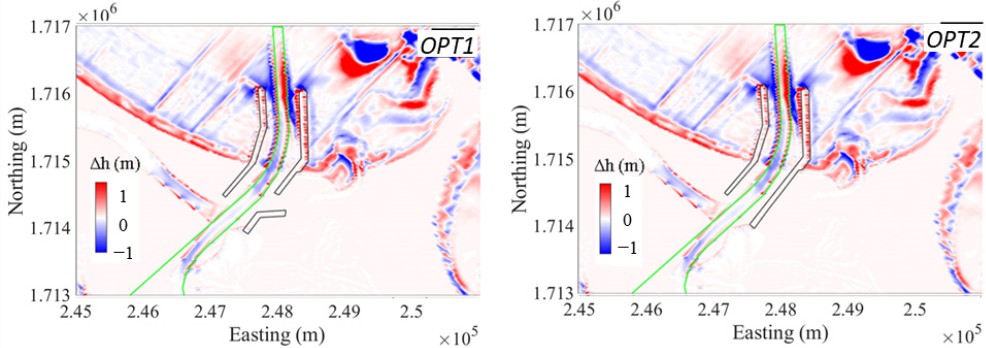

**Figure 19.** Bed level change in the area of Cua Lo after jetty construction.

### 4.1.2. Impact Assessment of Jetty Construction during Typhoon Molave

Figure 20 shows the largest wave field during typhoon Molave with the case of dredging and opening a new navigation channel according to OPT1 and OPT2 alternatives. The offshore wave height is high, reaching up to 5–6 m, and decreases to 2 m in the area of the Cua Lo inlet. The waves within the channel decrease significantly compared to current

condition, with the highest wave height at the top of the jetties being 0.25–1.0 m. However, the waves in the southern beach reach 2–3 m, causing instability and potentially impact on navigation and shipping.

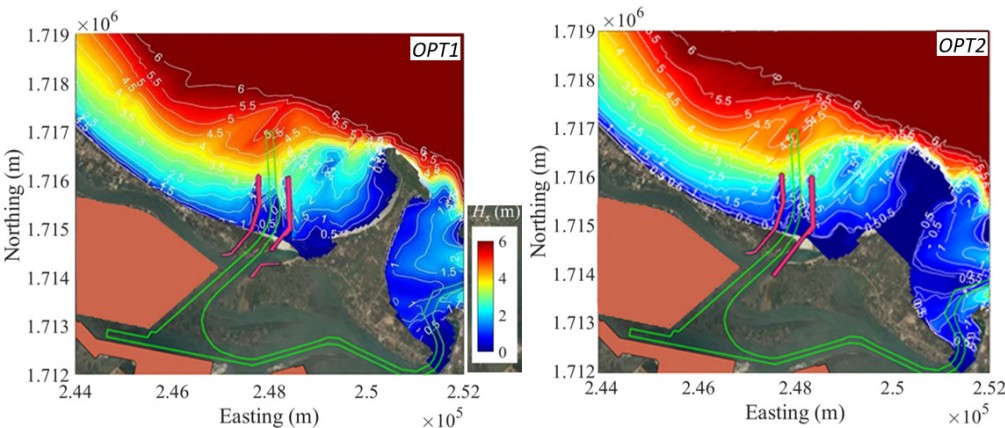

**Figure 20.** Wave field during typhoon Molave after jetty construction.

Figure 21 illustrates the velocity field during typhoon Molave after jetty construction. The flow is larger within the estuary and port area, with a velocity of 0.5–2.0 m/s in the new channel. The flow in OPT2 alternative is slightly higher than the one in OPT1 alternative.

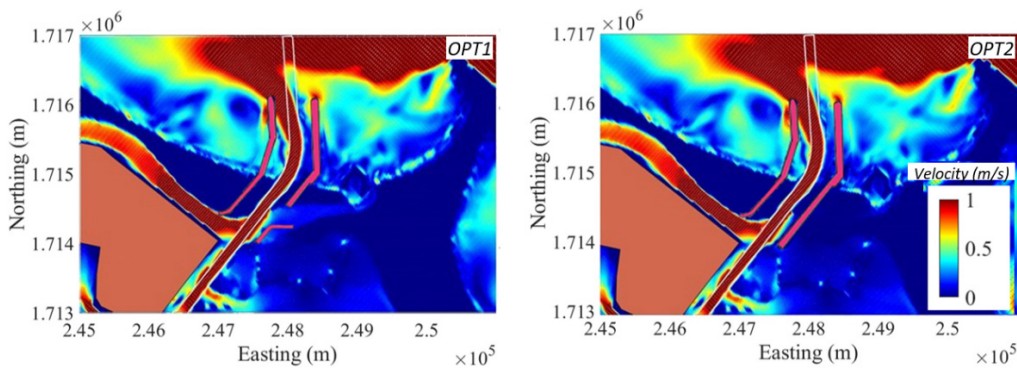

**Figure 21.** Current field during typhoon Molave after jetty construction.

4.1.3. Impact Assessment of Jetty Construction during the Flood Event

Figure 22 shows the maximum flow velocity during the flood event in November 2017 according to OPT1 and OPT2 alternatives. The flow velocity through the newly opened channel is very large with a maximum velocity of over 1.5 m/s. The flow range with high velocity from 1–1.6 m/s can reach 1.5–2 km outside the river mouth.

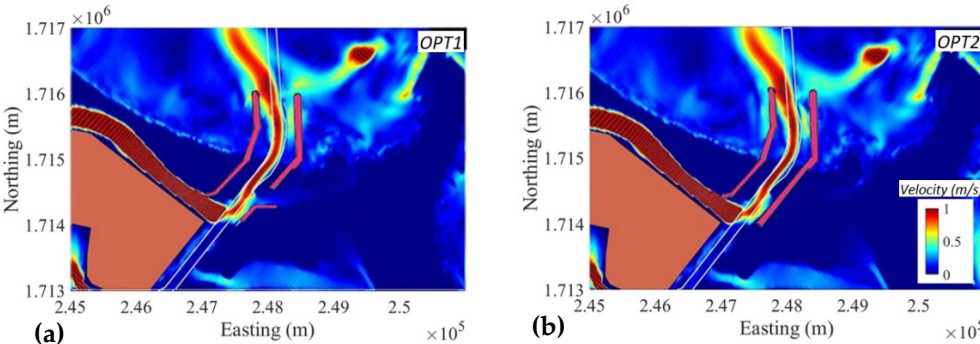

**Figure 22.** Current field during the flood with the frequency of 5% after jetty construction.

Figure 23 illustrates the bed level changes at Cua Lo estuary after the flood event. The section inside the river tends to be eroded 0.2–0.5 m; especially at the tip of the southern jetty where the erosion depth is approximately 0.5 m under both alternatives. The sand in this erosion area is then displaced and partly deposits within the channel and inlet area. The area of confluence between the Truong Giang river and the new channel is eroded and accreted by 0.2–0.5 m, and this amount of erosion is carried out by the river flow.

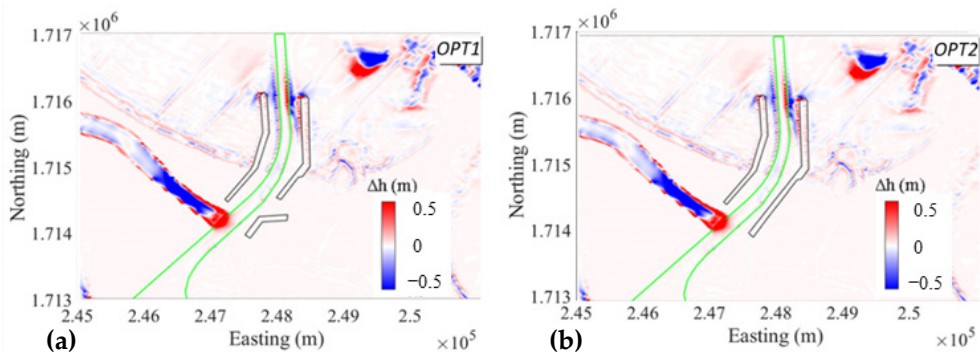

**Figure 23.** Bed level changes in the area of Cua Lo after the flood event and jetty construction.

Using simulations of wave fields and current flows within the study area under various weather conditions, including storms, monsoons, and floods, the authors have obtained a broad understanding of the changes in hydrodynamic factors and bed level change resulting from the proposed construction plan compared to the current conditions. To provide a more in-depth analysis of these changes, detailed investigations based on three evaluation criteria are presented hereafter.

- Evaluation of wave reduction effectiveness

To assess the wave transformation features of the construction alternatives in comparison to current conditions, simulated wave height at several points along the channel (presented in Figure 2) are extracted. Figure 24 shows that there is no significant change in the wave height at points from C8 to C13 during the Southeast monsoon compared to the current conditions. However, the maximum wave height decreased by 20–30% compared to the current condition during the storm, particularly at points from C8 to C13, which are located downstream of the navigation channel.

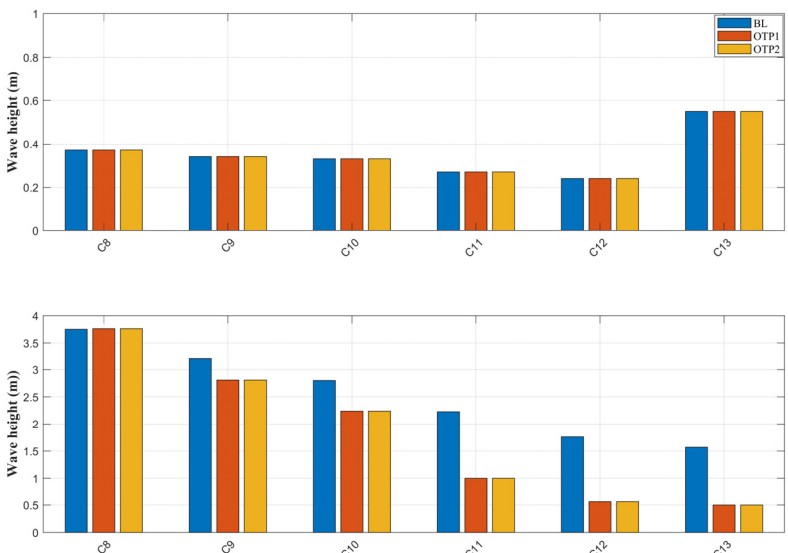

**Figure 24.** Wave height during the Southeast monsoon (**upper** panel) and the storm (**lower** panel).

- Evaluation of flood discharge effectiveness

In the natural condition, the flow velocity through the river mouth during the peak flood season is around 1.5 m/s. During the Northeast monsoon, flow velocities ranges from 0.5–1.0 m/s (Figure 25), while during the Southeast monsoon, velocities are smaller than 0.3 m/s. Thanks to the construction of the navigation channel, the efficiency of flood discharge into the sea is significantly improved with the maximum velocity through the new channel exceeding 1.5 m/s. Interestingly, despite being connected to the old Cua Lo river mouth, there is minor flow through this channel under OPT1. The old Cua Lo river mouth is expected to be a stable area due to its excellent protection from the new infrastructure and Ban Than cape. The flow velocity through the new channel was only slightly higher than that in the natural state.

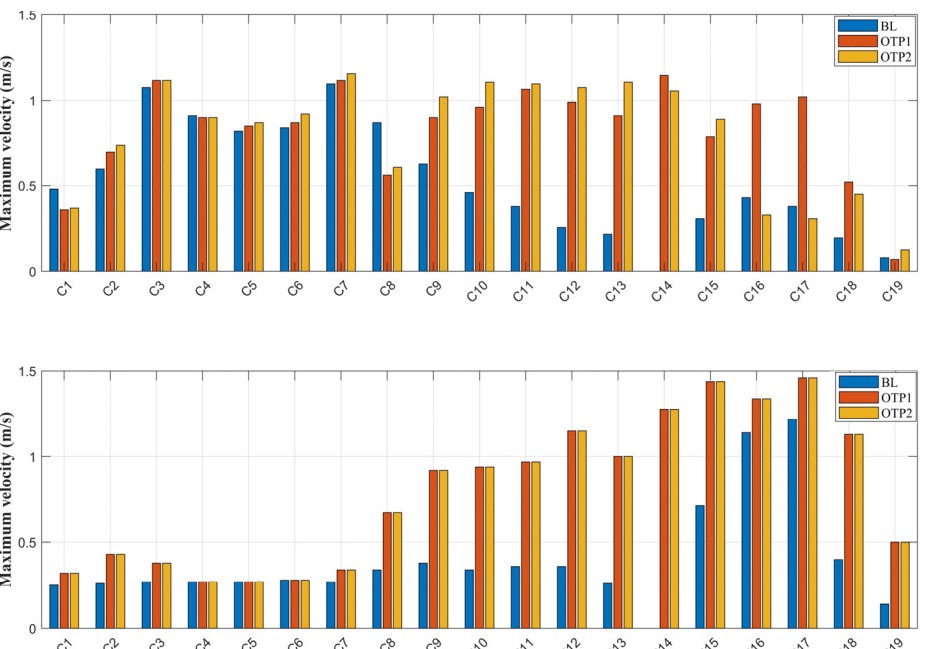

**Figure 25.** Maximum velocity during the Northeast monsoon (**upper** panel) and flood event (**lower** panel).

The velocity of water flow increased significantly between points C9 and C18 following the dredging of the channel and the opening of new river mouths under the OPT1 and OTP2 alternatives. This resulted in a maximum velocity of approximately 1.5 m/s during floods, indicating that river floods tend to discharge into the sea at a higher rate than the current condition (BL). It is worth noting that the flow velocity between points C9 and C18 was higher in the OTP2 alternative than in the OTP1 alternative. This suggests that flood discharge was more efficient when the new river mouth was not connected to the Truong Giang River via the old Cua Lo river mouth. Under OPT2 conditions, flood flows were more focused on discharging through the new river mouth than under OPT1 conditions. During storm conditions, the velocity of water flow between points C8 and C19 outside the sea increased under both OPT1 and OPT2 alternatives, with the highest velocity reaching over 1 m/s during the storm. However, the velocity of water flow between points C8 and C2 remained stable and unchanged compared to the BL condition. This indicates that the river gate adjustment was effective in stabilizing the water flow during a storm.

- Effectiveness of reducing sedimentation in the new Cua Lo channel

In order to assess the potential sedimentation and erosion phenomenon under the same hydrogeological conditions, 12 typical zones around the regulation works and Cua Lo estuary have been selected for further analysis (Table 3 and Figure 26a).

**Table 3.** Zoning of the areas for sedimentation and erosion analysis.

| TT | Zone | Description |
|----|------|-------------|
| 1 | R1 | The North bank zone of the navigation channel |
| 2 | R2 | The South bank zone of the navigation channel |
| 3 | R3–R9 | The zones within the navigation channel |
| 4 | R10 | The zone at the junction of Truong Giang river and navigation channel |
| 5 | R11 | The area of the port where ships turn around |
| 6 | R12 | The area within An Hoa lagoon |

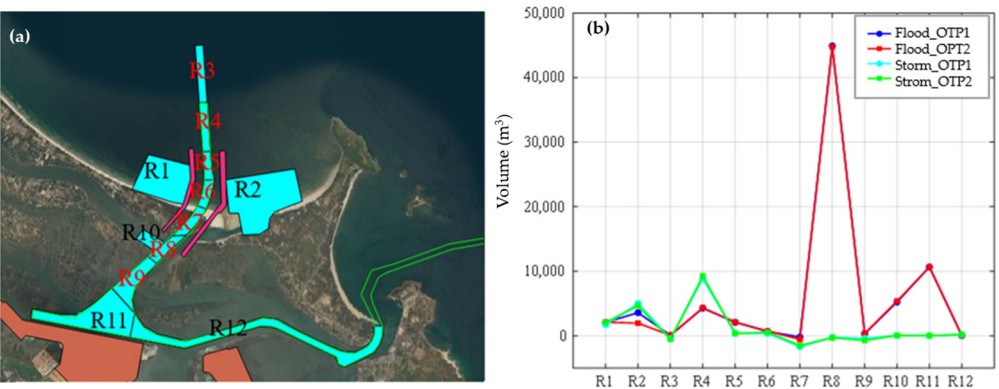

**Figure 26.** (**a**) Twelve zones for sedimentation and erosion analysis, (**b**) amount of sediment deposited under two alternatives.

Figure 26b shows that the differences in seabed terrain changes according to OPT1 and OTP2 alternatives are minor, both under storm and flood conditions. Overall, floods primarily cause sedimentation in the R8 area, resulting in a volume of approximately 45,000 m$^3$. Since this area is located at the confluence of the Truong Giang River and new navigation channel, sediment from upstream tends to accumulate at this zone. The total amount of sedimentation within the channel, i.e., the zones from R3 to R11, is about 68,000 m$^3$ under a typical flood event in both construction alternatives. Conversely, storms cause seabed terrain changes primarily in the mouth of the channel, spanning from zones R3 to R5, and result in a total sedimentation volume of approximately 9000 m$^3$.

The above analysis suggests that: (i) OPT2 alternative is more effective and practical than OPT1 alternative, and (ii) OPT2 guarantees better flood drainage and has a smaller amount of sediment deposited within the navigation channel.

*4.2. Assessment the Impact of Jetty Construction on Shoreline Evolutionary Trend along Quang Nam Coast*

This section focuses on investigating the shoreline evolutionary trend along 50 km sandy coast from Cua Dai estuary to Cua Lo estuary after the construction of two jetties in OPT2 alternative. The simulation results reveal the tendency of accretion in the northern part of Cua Lo after years of operation (Figures 27 and 28 and Table 4). Under the current situation, the amount of sediment eroded from the 15 km coastal segment, which starts at the location 20 km from the Cua Dai estuary, is the primary source of sediment that causes the elongation of sand spit and accretion in the areas in the northern part of Cua Lo and southern part of Cua Dai. This trend continues after the construction of regulation works at Cua Lo: an increase of erosion in the central segment of the coast (up to 50 m after 50-year of operation) and of accretion around both side of the coast. After 20-year and 50-year of operation, the northern coast adjacent to the Northern jetty reaches a distance of 250 m and 449 m towards the sea compared to the natural condition, respectively. This phenomenon stops only when sediment sources in the study area reach an equilibrium state (in the condition that there is no abnormal change of discharge at the estuary, bringing the sediment in and out of the estuary, causing changes in the sediment source). Since the sediment

is trapped in the left region of the Northern jetty, the construction of jetties prevents the elongation of the sand spit as well as the deposition within the navigation channel.

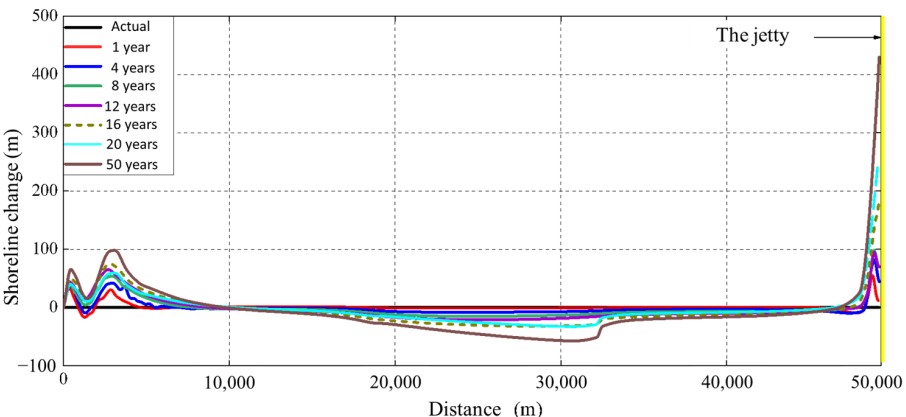

**Figure 27.** Shoreline change from Cua Dai estuary to Cua Lo estuary.

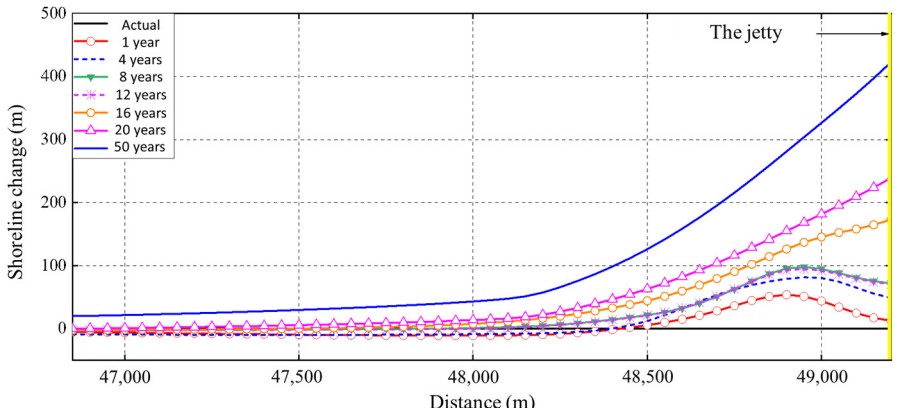

**Figure 28.** Shoreline change within 3 km in front of the Northern jetty.

**Table 4.** Statistical table of shoreline change (m/year) after construction.

| Location | Southern of Cua Dai | Middle Shoreline | Northern of Cua Lo |
|---|---|---|---|
| After 1 year | Max = +40.85 | Max = −0.29 | Max = +35.49 |
| | Min = −20.77 | Min = −2.05 | Min = −23.80 |
| After 4 years | Max = +45.30 | Max = −0.13 | Max = +77.07 |
| | Min = −7.58 | Min = −7.91 | Min = −17.95 |
| After 8 years | Max = +53.63 | Max = +0.94 | Max = +98.13 |
| | Min = 0.00 | Min = −15.17 | Min = −6.58 |
| After 12 years | Max = +65.00 | Max = +4.30 | Max = +98.48 |
| | Min = 0.00 | Min = −21.25 | Min = −5.84 |
| After 16 years | Max = +75.00 | Max = +8.57 | Max = +155.11 |
| | Min = 0.00 | Min = −32.17 | Min = −4.09 |
| After 20 years | Max = +76.00 | Max = +9.42 | Max = +252.17 |
| | Min = 0.00 | Min = −67.32 | Min = +6.1 |
| After 50 years | Max = +100.30 | Max = +12.56 | Max = +449.17 |
| | Min = 0.00 | Min = −83.81 | Min = +6.10 |

Note: The shoreline change has a negative value (−) means that the shoreline is eroding. Vice versa, the positive value (+) refers to accretion.

## 5. Conclusions

This study investigates the hydrodynamic and morpho-dynamic features in the area of Cua Lo estuary, both before and after port and navigation channel construction. The wave model with large domain and flow model with finer domain have been developed and well-calibrated, validated using Delft3D-WAVE and Delft3D-FLOW modules to analyze how these features vary under the impacts of the monsoon wave climate, typical storm, and river flood conditions. Additionally, the one-line shoreline model for Quang Nam coast is also developed using GenCade model with the optimal couple of empirical coefficients $K_1$ and $K_2$ are 0.4 and 0.2.

The simulation results reveal that the erosion occurs on the right bank of Cua Lo estuary, especially under flood condition with the maximum velocity exceeding 1.5 m/s and maximum depth of 0.5 m. Since the southern coast of Cua Lo is shielded by Ban Than rocky spit and coral reefs—which act as the no-transport boundary, the sediment transported along the coast from Cua Dai to Cua Lo in combination with the large amount of eroded sediment from the right bank are the sources for forming the sand spit that elongates toward the estuary.

The study evaluates the effectiveness of two design alternatives based on three evaluation criteria, i.e., wave reduction efficiency, flood discharge enhancement efficiency, and sedimentation reduction. During the storm event, the maximum wave height decreased by 20–30% compared to current condition, particularly at points from C8 to C13 that located at the end of the navigation channel. The construction of both OPT1 and OPT2 alternatives improves the efficiency of flood discharge into the sea, with the maximum velocity through the new channel exceeding 1.5 m/s. Under port construction conditions, the OPT2 is considered more effective and practical than OPT1 alternative since it guarantees better flood drainage and has a smaller amount of sediment deposited within the navigation channel.

The investigation of the shoreline evolutionary trend along 50 km sandy coast after the construction of two jetties under OPT2 alternative reveals that after 20-year and 50-year of operation, the northern coast adjacent to the Northern jetty reaches a distance of 250 m and 449 m towards the sea compared to the natural condition, respectively. This phenomenon stops only when sediment sources in the study area reach an equilibrium state. Since the sediment is trapped in the left region of the Northern jetty, the construction of jetties prevents the elongation of the sand spit as well as the deposition within the navigation channel.

**Author Contributions:** D.N.Q. and N.Q.D.A. wrote the first draft of the manuscript, performed the data collection and analysis, and prepared the figures and the literature review. H.S.T., N.X.T., H.T. and N.T.V. designed and contributed to interpreting the results. All authors have read and agreed to the published version of the manuscript.

**Funding:** This research is supported by "Development of model systems to assess and forecast morphological changes and countermeasures to stabilize the beaches in the Mid-Central Vietnam region" project (code: 42/22-ĐTĐL.CN-XNT), funded by the Ministry of Science and Technology (MOST), Vietnam.

**Institutional Review Board Statement:** Not applicable.

**Informed Consent Statement:** Not applicable.

**Data Availability Statement:** Not applicable.

**Conflicts of Interest:** The authors declare no conflict of interest.

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
