# Peer review of "Evaluation of Cua Lo Estuary’s Morpho-Dynamic Evolution and Its Impact on Port Planning"

_jmse, doi:10.3390/jmse11030611_

Round 1
Reviewer 1 Report
Comments:
1. English grammar should check carefully, e.g., the Abstract section should be revised as: “Coastal erosion and accretion along the Quang Nam coast in Vietnam have been increasing in recent years, causing negative impacts on the inhabitants and local ecology. The Cua Lo estuary in the Nui Thanh district has a complex hydrodynamic regime due to its connection to two estuaries and three different tributaries. Therefore, a detailed study of the mechanisms and processes of these phenomena is necessary to understand the potential impact of a proposed 50,000-ton cargo port. In this study, the Delft3D model is used to evaluate the morpho-dynamic changes in the area of Cua Lo under monsoon wave climate, storm, and flood conditions before and after port and navigation channel construction. Results show that without the port, the impacts of tidal currents and waves during monsoon storms lead to significant erosion on the south bank and accretion on the north bank. The GenCade model is also used to predict future shoreline changes after the construction of two jetties, revealing that after 50 years of operation, shoreline modifications will reach a distance of 449 m toward the sea compared to natural conditions. However, the design of the Northern jetty will ensure safety and proper operation without affecting the navigation channel.”
2. The Introduction section should consider more up-to-date studies related to the present study, e.g., Liang et al., 2022. On-Site Investigations of Coastal Erosion and Accretion for the Northeast of Taiwan. J. Mar. Sci. Eng. 2022, 10(2), 282.
3. What causes increasing coastal erosion and accretion in Quang Nam, Vietnam?
4. What is the complex hydrodynamic regime of the Cua Lo estuary?
5. What is the purpose of the detailed study of the mechanisms and processes of erosion and accretion in the Cua Lo estuary?
6. What is the Delft3D model, and how is it used in this study?
7. What are the impacts of tidal currents and waves during monsoon storms on the Cua Lo estuary without the proposed cargo port?
8. What is the GenCade model, and how is it used to predict future shoreline changes in the Cua Lo estuary?
9. How will the shoreline change after 50 years of operation of the proposed cargo port with two jetties?
10. How will the design of the Northern jetty ensure the safety and proper operation of the proposed cargo port?
Author Response
We would like to sincerely thank the anonymous reviewer for very constructive, valuable comments and suggestions on our paper. We greatly appreciate your valuable times and kind supports. Based on all the comments and suggestions, we are carefully considered and addressed your comments item-by-item. We believe that the quality of the revised manuscript has increased after addressing all of them. Please kindly find here for our reply to reviewer comments and the revised manuscript. All responses from authors are highlighted in “blue color”. In addition, the revised texts in the manuscript are also in “red color” for making easy to follow.

Reviewer 2 Report
These are the general comments. Please, see the attached PDF, where I made my comments directly in the text. Please, put attention to the improvement of the figures.
1. Descriptions of problems and tendencies of coastline changes are absent.
2. Too small information about the features of sediment transport in the study area is given.
3. The figures are overloaded by extra information, which has no link to the paper content.
4. At the same moment the details of new construction are not clear explain. For example, only after seen the Fig. 18 it became clear that the new artificial outlet will be constructed (digged) across the existed sandy spit
5. Obviously, the paper is based on deep engineering project. It is a good basis, but not enough for scientific paper. Some new knowledge should be demonstrated.
6. After reading the section 3.4 the problem of local shore development became little bit more clear. This section should be earlier in the paper structure.
7. In general, the paper is not focus to the one clear problem, too many additional information is given.
8. After presenting a lot of information, which show how deep and extended study was done, the final phase, the comparison of different alternatives OPT1 and OPT2, were presented very-very modest!
9. It seems that the differences between these alternatives have a small scale, while both are equal in terms of the influence to the general shoreline development (as on the figs. 26 and 27). Practically, it may be a first general conclusion. The second one should be based on comparison of these alternatives at the level of smaller scale.
10. I think the task to compare these alternatives should be based on formulation of clear criterions, but now these criterions were not presented in the paper. For example the OPT2 seems to be more attractive as it keeps the already existed stream (ecologically and socially valuable), but the erosion of right bank of this inlet probably will be not stopped, as the water flow will be not reduced too much.
11. I couldn’t support the publication of paper in the present form. The paper should be completely restructured. Scientific problem (for example, to explain the existed shoreline changes and to reveal the major forcing factors or events, which determine these changes) or the practical problem (how to stop the erosion but to keep some other important values ….) should be clear formulated.
12. The part of text, which is related to methods of modelling, calibration and validation is generally OK (see some comments in the text directly).
13. After formulation of the problem, the material related to it (avoid unnecessary details) should be presented in a traditional way (not as fragments of scientific report). Methods have to include the criteria of comparison or evaluating of alternatives. The results should present the direct application of all methods, including their evaluation using the formulated criteria. The conclusions should be more bright and show the benefit of the new construction in principle, and the better alternative for this construction.
14. The JMSE is an international journal, therefore authors should clear formulate something interesting for the broad international audience. Just now I ddin’t find anything like this.
15. I don't feel qualified to judge about the English language and style, but have the following comments:
- the text was completely understandable for me, I didn’t find any problem with language description
- I have a proposal the text be reviewed by an expert in English for punctuation and the use of articles.

Author Response

(The authors gave the same response as above.)

Round 2
Reviewer 1 Report
The authors have well addressed my comments and concerns. I suggest that this manuscript can be accepted for publication in the JMSE.
Reviewer 2 Report
I was satisfied with corrections maid by authors. Of course, it may be more and more improvement, but following th esentence that "best is the nemy of good" I would suggect to accept thу manuscript as it is.
Please, pay attention to figures 17 and 20. The fragments for OPT2 shows the flooding of the peninsula to east from the jetties. Once the wind influence is the same, is it correct? If it is correct, please, insert the explanation - why does the OPT2 configuration of jetties influence upstream the wave and water level field? Sorry, if I miss the explanation, if it is in the text.